# Deep Learning Model Compression and Hardware Acceleration for High-Performance Foreign Material Detection on Poultry Meat Using NIR Hyperspectral Imaging

**DOI:** 10.3390/s25030970

**Published:** 2025-02-06

**Authors:** Zirak Khan, Seung-Chul Yoon, Suchendra M. Bhandarkar

**Affiliations:** 1School of Computing, University of Georgia, Athens, GA 30602, USA; zirak.khan@uga.edu (Z.K.); suchi@uga.edu (S.M.B.); 2Quality and Safety Research Unit, U.S. National Poultry Research Center, U.S. Department of Agriculture—Agricultural Research Service, Athens, GA 30605, USA

**Keywords:** hyperspectral imaging, deep learning, inference optimization, post-training quantization, hardware acceleration, foreign material detection, poultry processing, industrial applications, real-time processing, sensor technology

## Abstract

Ensuring the safety and quality of poultry products requires efficient detection and removal of foreign materials during processing. Hyperspectral imaging (HSI) offers a non-invasive mechanism to capture detailed spatial and spectral information, enabling the discrimination of different types of contaminants from poultry muscle and non-muscle external tissues. When integrated with advanced deep learning (DL) models, HSI systems can achieve high accuracy in detecting foreign materials. However, the high dimensionality of HSI data, the computational complexity of DL models, and the high-paced nature of poultry processing environments pose challenges for real-time implementation in industrial settings, where the speed of imaging and decision-making is critical. In this study, we address these challenges by optimizing DL inference for HSI-based foreign material detection through a combination of post-training quantization and hardware acceleration techniques. We leveraged hardware acceleration utilizing the TensorRT module for NVIDIA GPU to enhance inference speed. Additionally, we applied half-precision (called FP16) post-training quantization to reduce the precision of model parameters, decreasing memory usage and computational requirements without any loss in model accuracy. We conducted simulations using two hypothetical hyperspectral line-scan cameras to evaluate the feasibility of real-time detection in industrial conditions. The simulation results demonstrated that our optimized models could achieve inference times compatible with the line speeds of poultry processing lines between 140 and 250 birds per minute, indicating the potential for real-time deployment. Specifically, the proposed inference method, optimized through hardware acceleration and model compression, achieved reductions in inference time of up to five times compared to unoptimized, traditional GPU-based inference. In addition, it resulted in a 50% decrease in model size while maintaining high detection accuracy that was also comparable to the original model. Our findings suggest that the integration of post-training quantization and hardware acceleration is an effective strategy for overcoming the computational bottlenecks associated with DL inference on HSI data.

## 1. Introduction

Ensuring the safety and quality of poultry products is of high importance in the poultry industry, as contamination with foreign materials (FMs) poses severe health risks to consumers and can lead to high economic losses to the industry due to product recalls [1]. Non-biological FMs, such as rubber, plastic, and metal fragments, are common contaminants introduced inadvertently during the deboning, trimming, portioning, and packaging processes in poultry processing plants [2,3]. The efficient detection and removal of these FMs are critical to maintaining food safety standards and complying with regulatory requirements.

While many traditional detection techniques, including X-ray imaging, metal detection, thermal imaging, spectroscopy, and conventional color imaging, have been employed to identify FMs in food products [4], each presents inherent limitations: X-ray imaging, though highly accurate for dense objects, struggles with low-density materials like plastic and rubber; thermal imaging is sensitive to environmental fluctuations; terahertz spectroscopy is hindered by the high water content in poultry products; and color imaging often fails to distinguish contaminants from meat due to similar appearances [5,6,7,8]. Hyperspectral imaging (HSI) has emerged as a powerful non-destructive sensing technology that combines imaging and spectroscopy to capture detailed spatial and spectral information from materials [9]. When integrated with advanced deep learning (DL) techniques, such as convolutional neural network (CNN) and generative adversarial network (GAN), these HSI-based DL solutions further enhanced the detection capabilities in the remote sensing and food safety sectors [10,11]. This fusion allows for the extraction of deep features from high-dimensional HSI data, leading to improved accuracy in classification and detection tasks. For instance, Chung et al. [12] demonstrated that a fusion model combining two HSI modalities—visible near-infrared (VNIR) HSI and short-wave infrared (SWIR) HSI—achieved superior precision and recall in detecting FMs on broiler meat compared to using standalone modalities. Similarly, Seo et al. [13] showed that VNIR HSI coupled with a 1D CNN model outperformed traditional support vector machine (SVM) methods in detecting organic residues on food machinery surfaces. Additionally, Yang et al. [14] highlighted the effectiveness of CNNs and sparse autoencoders in identifying mold in stored maize kernels using HSI data. Furthermore, Campos et al. [10] presented a semi-supervised DL approach leveraging near-infrared (NIR) HSI and GANs to detect FMs on poultry meat. These studies underscore that the integration of HSI and DL techniques presents a more robust approach for enhancing foreign material (FM) detection in food processing, offering great improvements over traditional methods.

However, modern poultry processing lines operate at high speeds, processing between 140 (a typical speed in the U.S.) and 250 birds per minute (BPM) [15], which presents deployment challenges for HSI-based DL models in such high-throughput environments. This is mainly due to the high-dimensional nature of HSI data, the over-parameterized DL models, and the resource-constrained environment. The high volume of input HSI data for these advanced DL models increases the required intermediate activation memory and computation demands during inference. Similarly, the large number of pre-trained models’ weights also increases the required memory footprint and computational needs [16]. To avoid such memory and computation bottlenecks in high-throughput and resource-constrained environments like food processing lines, efficient DL- and HSI-based solutions are needed. In recent years, several studies have been dedicated to improving DL model efficiency in the computer vision domain. Model compression and hardware acceleration techniques provide efficient DL models without compromising accuracy.

Model compression aims to reduce the size and computational demands of DL models, facilitating faster inference and lower memory usage [17,18,19,20,21]. Two primary model compression methods are pruning and quantization [22,23]. Pruning involves reducing the size of pre-trained DL models by eliminating redundant or less significant weights and activations while maintaining accuracy [24]. Pruning can be both fine-grained and coarse-grained. Both approaches have certain merits and drawbacks. The fine-grained approach allows for larger compression ratios but is challenging to accelerate hardware due to the complexity of sparse matrix multiplication. In contrast, the coarse-grained approach offers moderate compression rates but is easier to accelerate because it is similar to running a smaller model [25,26]. Moreover, pruning typically requires retraining or fine-tuning the pruned model to recover any accuracy loss [25]. The size of a DL model depends on the number of weights and the bit width, also known as precision in graphics processing unit (GPU) computing, required to store each weight. Quantization involves reducing the bit width to store each weight. Because low-bit operations not only reduce memory footprint but are also cheaper both from arithmetic operations and data movement perspectives [27,28,29]. Some commonly supported low-bit data types by different hardware architectures are 32-bit floating-point (FP32), 16-bit floating-point (FP16), and 8-bit integer (INT8). Quantization can be applied during training, allowing the model to adjust weights to reduced precision, called Quantization Aware Training (QAT). QAT generally achieves better accuracy with low bit widths but requires access to the original data and is best-suitable when training a model from scratch [30]. On the other hand, post-training quantization (PTQ) applies quantization to a pre-trained DL model without the need for retraining [31]. This process primarily involves mapping floating-point weights and activations into discrete levels using scaling factors (S) and zero points (Z). PTQ can quantize an entire tensor with a single scale (per-tensor) or assign unique scales per channel (per-channel), creating a balance between computational overhead and accuracy [28,32]. More advanced techniques, such as group quantization, refine the scaling factors at an intermediate level, optimizing both memory usage and performance [33]. Additionally, a small calibration set is often used to determine the dynamic range of activations (rmin, rmax), which helps reduce quantization errors near the centers of distribution [34,35,36]. However, when quantizing weights and activations directly from FP32 to FP16, it is not typically necessary to calculate optimal quantization parameters like zero points and scales using a calibration dataset, making this process more convenient [35,36]. Calibration is mainly required when quantizing activations and weights to integer formats, such as INT8. In this study, we adopted FP16 PTQ to eliminate the need for retraining and avoid the requirement for a calibration dataset.

Hardware acceleration exploits specialized processors to enhance the computational efficiency of DL models. GPUs offer high parallelism, making them well-suited for the matrix operations prevalent in DL inference. Field-programmable gate arrays (FPGAs) allow for custom hardware configurations, enabling tailored acceleration for specific models or operations. In this study, we accelerated the HSI-based DL model on a GPU that is based on Ampere Architecture supporting only FP16 and INT8 quantization.

Although the effectiveness of the aforementioned optimization techniques is recognized in various fields, their use in HSI-based DL models remains underexplored. This is particularly evident when it comes to achieving real-time inference with HSI-based DL models for FM detection in poultry meat. Research was needed to address this challenge. In this study, we explored PTQ and hardware acceleration to optimize a semi-supervised FM detection DL model with two pre-trained GAN discriminators, for high-performance FM detection using HSI data. Our contributions are as follows: (1) Hardware acceleration with TensorRT: We developed two inference engines for the two discriminators utilizing NVIDIA’s TensorRT to accelerate model inference, achieving a fivefold reduction in inference time compared to unoptimized GPU-based inference and exceeding the inference-time requirements of poultry processing. (2) Model compression using PTQ on TensorRT without fine-tuning: We applied FP16 PTQ to quantize the weights and activations of the same pre-trained semi-supervised FM detection DL model with two GAN discriminators. This process did not require calibration data and any fine-tuning, while still maintaining the accuracy of the pre-trained model. As a result, we achieved a 50% reduction in model size. (3) We conducted simulations using two hypothetical hyperspectral line-scan cameras to evaluate the feasibility of deploying an optimized HSI-based DL model for the real-time detection of FMs.

## 2. Materials and Methods

### 2.1. Dataset and Imaging System

The dataset employed in this study consisted of hyperspectral images captured from 12 boneless, skinless broiler breast fillets (pectoralis major muscle) using a push-broom line-scanning SWIR HSI system (Hyperspec^®^ SWIR, Headwall Photonics, Fitchburg, MA, USA) with a wavelength range from 1000 to 2500 nm. The HSI system was equipped with a mercury–cadmium–telluride (MCT) camera and a spectrograph and operated via custom C++ software with tungsten halogen illuminators. Intensity calibration and denoising techniques were applied to optimize image quality for further analysis, with additional procedural details also available in our previous work [10,12].

Utilizing the SWIR HSI system, each fillet was imaged under two conditions: clean without FMs first and contaminated with FMs for later imaging. The contamination involved 30 distinct types of FMs commonly found in poultry processing environments, such as polymers, metals, and wood, detailed in the prior work [10,12]. The target sizes of FMs were approximately 2 × 2 mm (small) and 5 × 5 mm (large). Each fillet for imaging was placed on a blue food-grade polyurethane conveyor belt background. In total, 48 SWIR hyperspectral images were collected: 24 clean and 24 contaminated (12 with each size of FM). These SWIR hyperspectral images (1000–2500 nm) were subsequently transformed into NIR hyperspectral images with 95 spectral bands from 1000 to 1700 nm. Figure 1 shows three exemplar RGB pseudo-color images derived from clean and contaminated NIR hyperspectral images utilized in this study.

These NIR hyperspectral images were employed in this study to evaluate the performance of a pre-trained semi-supervised DL model with two GAN discriminators for FM detection on the surface of poultry meat [10]. This DL model was originally trained on part of the dataset; however, in this study, all images in the dataset were used to compare and evaluate inference performance—such as accuracy, inference time, and throughput—among different models, including the original DL model on CPU and GPU platforms and the optimized DL model through the proposed PTQ and hardware acceleration techniques. This analysis aimed to assess the effects of optimization techniques on model performance in real-world scenarios, specifically simulating the typical imaging conditions for modern poultry processing lines.

### 2.2. Inference Pipeline for FM Detection

The inference pipeline for FM detection includes several key components: pre-processing steps applied to the input hyperspectral data, an inference model based on a semi-supervised DL model with two GAN discriminators [10], an inference engine comprising two sub-engines that executes inputs using the model, and post-processing steps to generate the prediction output. The pipeline and data flow are illustrated in Figure 2. Pre-processing steps involve calibrating raw hyperspectral data to relative reflectance values and reshaping the calibrated hyperspectral image (a 3D data cube) into a 2D array of spectral vectors. The DL model with two GAN modules was uniquely designed to address the spectral heterogeneity found in two major tissue types of skinless poultry meat, specifically in distinguishing the hyperspectral signatures of muscle tissue and non-muscle external tissues, such as external fat and connective tissues, observed on the surface. The model operates on the hypothesis that training with clean chicken muscle and non-muscle external tissues’ spectral responses. Thus, two GANs are sufficient for the model to recognize typical spectral characteristics of normal chicken breast meat. Once the model is trained, any anomalous spectral signatures lead to significantly higher prediction errors during inference, indicating the presence of FMs. The two GAN discriminators in the pre-trained inference model were frozen and saved as HDF5 files. These HDF5 files were then used for unoptimized inference engines, such as Tensorflow-CPU and Tensorflow-GPU, and optimized inference engines with TensorRT GPU. The saved two GAN discriminator files were converted to ONNX format for compatibility with TensorRT. Given a hyperspectral image, both inference sub-engines received the same pre-processed spectral vectors for inferencing and outputting the probabilities of FM presence at each pixel independently. Post-processing steps include thresholding to generate binary output vectors, reshaping the output binary vectors to 2D images representing the original spatial resolution, median filtering for denoising, creating individual detection maps for each engine, and combining the individual detection maps with a data fusion rule for the final pixel-level FM detection map. The post-processing steps were performed offline on the CPU.

#### 2.2.1. Inference Model

As mentioned above, the inference model employs two GAN discriminators, designed to process distinct spectral responses associated with muscle and non-muscle external tissues in chicken breast fillets separately. This specialized approach enhances the precision of FM detection within hyperspectral data. Each GAN module for training consists of two competing components: the generator and the discriminator.
**Generator:** The generator includes several 1D convolutional layers activated by ReLu, leading to a sigmoid activation layer at the end. The generators in two GAN models create synthetic spectral data that closely resemble clean (normal) hyperspectral data for each tissue type, respectively. The generator was only used during model training in our case.**Discriminator:** The discriminator adopts a 1D U-Net architecture, consisting of an encoder and a decoder, to process spectral vectors at individual pixel locations (x,y). Before being sent to two GAN modules, the input spectral vectors are initially subjected to clustering into two groups using a 1D Gaussian Mixture Model (GMM). This clustering segregates the spectral data into distinct clusters representing muscle and non-muscle external tissues, which are then processed by their respective discriminators for further analysis. Note that this GMM module for spectral clustering is only necessary during the model training phase. During the inference, only the encoder part of the two discriminators is utilized. Therefore, the inference model consists of two pre-trained GAN discriminator encoders. The overall architecture of the discriminator is shown in Figure 3. The discriminator outputs probabilities that are thresholded and further processed for binary classification at each pixel, indicating the presence (1) or absence (0) of FM, as previously described. For the GAN discriminators, a dummy band was added to each spectrum, resulting in a final spectrum size of 96.

#### 2.2.2. Creation of Pre-Trained Inference Model

The process of creating a pre-trained inference model [10] for the study is briefly described next. The DL model with two GAN modules was trained using approximately 850,000 spectral responses from a subset of 12 clean hyperspectral images in the dataset. This training phase focused on enabling the two GAN models to discern between normal (muscle and non-muscle) and anomalous spectral patterns of any FMs effectively. Following training, the inference model with two GAN discriminators was tested using approximately 2.6 million spectral responses, as independent test samples. This testing phase assessed the model’s accuracy. The model was developed using TensorFlow 2.12.0 and Python 3.10.10. The trained two GAN discriminators were saved as HDF5 files.

### 2.3. Optimizatized Inference

The optimization process involved hardware acceleration and model compression (PTQ) through NVIDIA’s TensorRT, a high-performance DL framework designed to improve model inference on NVIDIA GPUs. The inference model was optimized using hardware acceleration and PTQ to improve performance for real-time HSI operations.

#### 2.3.1. Hardware Acceleration

The optimization process began by freezing the pre-trained inference model and converting its format from TensorFlow to ONNX suitable for NVIDIA’s TensorRT. The DL inference optimization framework within TensorRT is specifically designed to enhance performance on NVIDIA GPUs by creating GPU-optimized inference engines. The models benefit from various features, including dynamic tensor memory and multi-stream execution. Dynamic tensor memory efficiently manages memory allocation, reducing energy consumption and minimizing operational overhead by allocating memory to tensors only when needed. In contrast, multi-stream execution facilitates the parallel processing of multiple data inputs, significantly enhancing performance by delivering low latency and high throughput.

The construction of a TensorRT inference engine involved several important steps. These steps include creating a network definition, importing the inference model with two GAN discriminators using the ONNX parser, and creating a TensorRT engine with two sub-engines, one for each discriminator, utilizing a builder mechanism. The process of inference in a TensorRT runtime consists of six key steps: (1) initiating an inference execution context; (2) allocating memory on the CUDA device for both input and output data; (3) transferring input data from the host to the allocated input memory on the CUDA device; (4) executing inference through the TensorRT engines using an API that supports the asynchronous execute for improved performance; (5) moving the output data back to host memory; (6) synchronizing the stream used for data transfers and inference execution to ensure the completion of all operations.

In our initial hardware acceleration setup via TensorRT, the model remains in FP32 precision. This primarily engages CUDA cores rather than leveraging half-precision Tensor cores. Although we achieve speedups over CPU and unoptimized GPU inference, due to optimized kernel fusion, dynamic memory allocation, and multi-stream parallelism, the full benefits of hardware acceleration are only realized after compressing the model to FP16.

#### 2.3.2. Model Compression

As a model compression strategy, the inference model with two pre-trained GAN discriminators underwent PTQ as its first step. This process reduced the precision of the model’s weights and activations from FP32 to FP16. The goal of this reduction was to decrease computational demands and enhance memory efficiency while maintaining model accuracy without any fine-tuning. As a result of PTQ, the model size became smaller. After quantization, the inference model with two GAN discriminators was accelerated using TensorRT.

#### 2.3.3. Computational Complexity

Each discriminator model for inference consists of 1D convolutional layers that progressively increase the number of channels (2, 4, 8, 16, 32) with stride-2 down-sampling in selected layers. The inference model comprises approximately 6359 parameters and 36,000 multiply-accumulate (MAC) operations per forward pass (see Section 2.2.1 for detailed layer specifications). When quantizing a model from FP32 to FP16, the number of MAC operations in the inference model stays the same. However, the arithmetic cost per MAC is reduced because fewer bits are used. In addition, the storage required for parameters is effectively halved, decreasing from approximately 24.8 KB to around 12.4 KB. This reduction helps us to alleviate memory bandwidth demands.

Once quantized, Tensor cores on NVIDIA GPUs process FP16 data at higher throughput than FP32 computations using CUDA cores, often achieving 2 to 4 times the theoretical speedups in terms of floating-point operations per second (FLOPS) [37,38]. By reducing memory usage, decreasing the bit-width per operation, and utilizing specialized hardware blocks (Tensor cores) for half-precision calculations, FP16 PTQ allows for faster end-to-end inference without the need for model retraining. This combination of model compression and hardware acceleration is central to achieving real-time speeds in our hyperspectral detection pipeline.

### 2.4. Experimental Setup and Conditions

#### 2.4.1. Experimental Setup

The experimental setup for evaluating the performance of unoptimized and optimized inference models was simulated by replicating real-world, real-time HSI conditions for FM detection in poultry processing. The dataset used in this study was derived from a comprehensive dataset of approximately 3.4 million spectral responses (96 bands per spectrum in the range of 1000–1700 nm) obtained from 48 hyperspectral images used in the previous study [10]. This dataset was sampled or expanded to create the necessary datasets for various experimental conditions, requiring only a set of input spectral vectors as input and a set of output vectors.

For the simulation, two hypothetical hyperspectral cameras were modeled after two commercial products [39]: an SWIR hyperspectral camera (Hyperspec^®^ SWIR, Headwall Photonics, Fitchburg, MA, USA) and a NIR hyperspectral camera (Specim FX17, Specim, Oulu, Finland). The SWIR camera has an MCT detector with 320 spatial pixels × 256 spectral bands in the wavelength range of 1000–2500 nm, while the NIR camera has an InGaAs detector with 640 spatial pixels × 224 spectral bands in the wavelength range of 900–1700 nm. Halved broiler breast fillets vary in size, measuring up to 20.33 cm (length) × 12.47 cm (width) in published studies [40,41]. In this study, we hypothesized that the fillet dimensions would be 9 inches (22.86 cm) in length and 5 inches (12.7 cm) in width, representing an upper bound. Two different imaging conditions were established: one for the SWIR camera where the longitudinal direction of a fillet was perpendicular to the direction of the image scan line, and another for the NIR camera where it was parallel to the image scan line. The NIR camera was assumed to produce a hyperspectral image of 640 pixels (image width, pixel resolution of 0.4 mm) by 168 scanlines (image height, pixel resolution of 0.8 mm), while the SWIR camera was assumed to produce a hyperspectral image of 320 pixels (image width, pixel resolution of 0.4 mm) by 800 scanlines (image height, pixel resolution of 0.3 mm). It was assumed that both cameras would produce 96 wavebands including a dummy band ranging from 1000 to 1700 nm to align with the input dimensions required for the pre-trained inference model. Therefore, the dimensions of the hypercubes studied were 320 width × 800 height × 96 bands for the SWIR camera (256,000 spectra) and 640 width × 168 height × 96 bands for the NIR camera (107,520 spectra). Notably, the maximum frame rates of the two cameras were 45 frames per second (FPS) and 670 FPS, corresponding to lines per second in hyperspectral imaging.

#### 2.4.2. Experimental Conditions

Key experimental conditions included four inference engines, each with two sub-engines, E1 and E2, corresponding to the two GAN discriminators D1 and D2, varying batch sizes, and diverse simulation setups. Four different inference engines—TensorFlow-CPU, TensorFlow-GPU, TensorRT-FP32, and TensorRT-FP16—were compared. A batch size refers to the number of input spectral vectors processed simultaneously during the inference runtime. As aforementioned, two hypothetical hyperspectral cameras were used in the study.

Inference Engines: Four different inference engines, each with two sub-engines, were established to measure the variations in model performance under different simulation conditions. This study compares the performance of these inference engines to find which engine performs best in different scenarios.

**TensorFlow-CPU:** This inference engine (two sub-engines) loads and executes the pre-trained inference model (two GAN discriminators) on the CPU without further optimizations using the TensorFlow CPU version. This inference engine provides a baseline benchmark for performance under highly resource-constrained conditions, such as an embedded industrial computer. It serves as a reference point for subsequent optimizations. The model’s structural or computational specifics did not change.**TensorFlow-GPU:** The inference model was loaded and executed on a single GPU board. This inference engine was utilized to directly evaluate the enhancements achievable through GPU acceleration alone, using the TensorFlow GPU version without further optimizations or modifications similar to TensorFlow-CPU.**TensorRT-GPU (FP32):** The two discriminators for inference, originally as files in the HDF5 format, were converted to the ONNX format to ensure compatibility with NVIDIA’s TensorRT. The ONNX files were then loaded and executed on the GPU through TensorRT. This inference engine took full advantage of TensorRT’s capabilities to enhance the execution efficiency of the models while maintaining their original FP32 precision for weights and activations. Thus, the focus was on improving performance through hardware acceleration.**TensorRT-GPU-PTQ (FP16):** This inference engine is essentially the same as TensorRT-GPU except for the added PTQ with TensorRT optimization. Initially, the model’s weights and activations were quantized from FP32 to FP16 (half-precision), aiming to reduce computational demands and enhance memory efficiency. Subsequently, these quantized models were accelerated using TensorRT to maximize the computational performance, assessing the overall impact of both quantization and advanced hardware acceleration on the models’ effectiveness.

Batch Size: A batch size in the number of spectral vectors (i.e., pixels) cannot be generalized for all scenarios, as different hyperspectral cameras and imaging conditions can result in varying wavelengths per pixel and thus require different memory sizes. For example, a batch size of one spectrum with 100 wavebands requires the same amount of memory as a batch size of 10 spectra with 10 wavebands per spectrum. However, batch sizes differ in these two cases (1 versus 10). To standardize the batch size and aid the understanding of readers, we also provide a batch size in memory size, measured in bytes, such as kilobytes (KB), megabytes (MB), and gigabytes (GB). The batch sizes in bytes were first designed to increase approximately by powers of 10, specifically: 1 KB, 10 KB, 100 KB, 1 MB, 10 MB, 100 MB, and 1 GB. The batch size in the number of pixels was then obtained by calculating the nearest integer using the formula, (batch size in KB)/(96 × 4 bytes). For example, the batch size of 1 KB contained 3 spectral vectors with 96 bands of 32-bit floating numbers. Note that the number of spectral vectors was provided to an inference engine as a batch size.Static and Dynamic Batch Size: When setting the input shape in TensorRT, a batch size affects performance. Static and dynamic batch sizes are compared in this study. A static batch size refers to a fixed one that does not change at runtime, whereas a dynamic batch size is a variable batch size that can change at runtime, providing flexibility to handle varying batch sizes. The static batch sizes used are 1 KB, 10 KB, 100 KB, 1 MB, 10 MB, 100 MB, and 1 GB. The dynamic batch size was defined using three parameters: minimum, optimal, and maximum values. An inference engine configured with a dynamic batch size works for all batch sizes within the [minimum, maximum] range and is optimized for performance at an optimal size within the range. The minimum size ensures responsiveness and quick processing for smaller data inputs, the optimal size provides the best performance in throughput and resource utilization, and the maximum sets the engine’s upper limit to manage the largest expected data volumes efficiently. In the study, the static batch sizes were set as the optimal sizes, and the ranges were defined as follows, [1, 10], [1, 50], [1, 500], [1, 5000], [1, 50,000], [1, 500,000], and [1, 5,000,000] (the number of pixels). A comparative experiment was designed to evaluate the performance of the static and dynamic batch sizes.

### 2.5. Computing Environment

This subsection outlines the computing infrastructure used for the experiments, which is critical for optimizing and evaluating the performance of the inference models in real-time scenarios. The hardware configuration included a Lenovo IdeaCenter 5i desktop PC (Lenovo Group Limited, Morrisville, NC, USA) equipped with a 12th generation Intel(R) Core(TM) i7 processor at 2.10 GHz and an NVIDIA GeForce RTX 3060 GPU, which has 12 GB of memory and 3584 CUDA cores. This setup provided sufficient computational power and memory capacity necessary for handling the demanding tasks of DL and real-time data processing required by the study. For software, Python was chosen due to its robust ecosystem and flexibility, crucial for high-performance computational tasks in HSI. The TensorRT Python API played a pivotal role in optimizing DL models for inference on NVIDIA GPUs, enhancing computational efficiency significantly. The specific versions of the software tools employed included CUDA 12.1.1, which maximizes GPU resource utilization, TensorFlow 2.13.1 for conducting baseline tests, and TensorRT 8.6.1 for the optimization and acceleration of the models.

### 2.6. Performance Evaluation

This subsection elaborates on the metrics utilized to evaluate the performance of four inference engines, focusing on accuracy, latency time, and throughputs to quantify the enhancements in real-time FM detection capabilities. These metrics allow for a thorough comparison of performance before and after optimization. The insights gained from these evaluations are crucial in validating the effectiveness of hardware acceleration and model compression.

#### 2.6.1. Accuracy

The accuracy of FM detection at the pixel level was assessed using precision, recall, F1 score, balanced accuracy (BACC), and overall accuracy (ACC). A binary abnormal pixel prediction (indicative of FM or contamination) was considered a positive outcome, while a binary normal pixel prediction (clean) served as a negative outcome. True positives (TPs) represented the number of correctly predicted FM pixels, and true negatives (TNs) indicated the number of correctly identified normal pixels. False positives (FPs) were the number of normal pixels misclassified as FM, and false negatives (FNs) were the number of FM pixels misclassified as normal. Precision, defined as the ratio of correct FM predictions to all predictions labeled as FM, and recall, or sensitivity, measured as the ratio of correct FM predictions to the actual number of FM pixels, were calculated. The F1 score, a harmonic mean of precision and recall, ranged from 0 (indicating poor performance) to 1 (indicating perfect accuracy), calculated without considering the TN due to the significant class imbalance in the dataset. Balanced accuracy (BACC) took into account this imbalance, averaging the sensitivity and specificity (true negative rate), with values ranging from 0 to 1.

#### 2.6.2. Latency Time

Inference latency time was measured as the end-to-end execution time of the engine’s forward inference cycle, excluding data retrieval, engine initialization, and input preprocessing. To ensure an accurate measurement of inference time, five preliminary “warm-up” operations were conducted. Following these operations, the execution time was recorded over 10 repetitions to compute a robust mean inference time. Time measurements were made using the “time” function from Python’s “time” module, capturing timestamps before and after inference execution. Latency time is also called inference time interchangeably in this paper.

#### 2.6.3. Throughput (MB/s)

Inference throughput, measured in MB/s, indicates the amount of data being inferred per second. This measure is calculated by dividing the total amount of inferred data in bytes by the total latency time. As previously mentioned, in our experiments, the batch size began at 1 KB of hyperspectral data, which is approximately equivalent to three pixels, and increased tenfold, until reaching the maximum limit of 1 GB that the GPU could handle without memory issues. It is important to note that the inference engines using TensorRT were specifically built for each batch size—whether static or dynamic—by utilizing CUDA kernels optimized for those sizes. In contrast, TensorFlow-CPU and TensorFlow-GPU engines operated with a static batch size at runtime. The effects of batch size and dynamic batch size over static batch size were analyzed with mean inference time per batch, standard deviation of inference times, and inference throughput.

#### 2.6.4. Hypercube Rate

To better align with the operational demands of poultry processing lines, we evaluated another throughput in terms of the number of hyperspectral images processed per second, called a hypercube rate. This metric also implies how many broiler breast fillets can be inspected per second using the proposed methods. In typical large U.S. poultry processing plants, the evisceration lines operate at 140 BPM, which produce 280 halved breast fillets per minute and equivalently 4.7 halved fillets per second at the downstream lines. The maximum line speed for the poultry industry can reach up to 250 BPM, equating to 8.3 halved fillets per second, in other countries that allow higher line speeds. Thus, it is critical for a vision system to keep up with this speed requirement.

The calculation of the hypercube rate was based on the throughput rate (measured in batches per second or MB/s) and the batch size (measured in pixels per batch or KB). The hyperspectral imaging conditions configured with the two hypothetical NIR and SWIR hyperspectral cameras (see Section 2.4.1) required 107,520 and 256,000 pixels, respectively, to capture a hyperspectral image (i.e., a hypercube) with the camera’s maximum spatial dimensions for scanning a halved broiler breast fillet entirely. The SWIR camera can be described as a 256k-pixel SWIR camera with 96 bands, while the NIR camera can be referred to as a 108k-pixel NIR camera with 96 bands. By relating the throughput to these pixel requirements, we established a metric that provides a realistic gauge of performance across different operational scales and batch sizes for each inference engine examined.

## 3. Results and Discussion

### 3.1. Inference Accuracy

Table 1 presents a detailed breakdown of the pixel-level detection accuracies of four inference engines, each with two sub-engines used in this study. The table presents each sub-engine’s results as well as the final fused results. In the table, the muscle channel and non-muscle channel refer to sub-engines E1 and E2, which utilize discriminators D1 and D2, respectively. The “fused channels” rows represent the pixel classification results of the final fused inference engine’s results, predicted from both muscle and non-muscle channels and appropriate post-processing steps described in Section 2.2 and Figure 2. The “fused channels” rows provide a comprehensive assessment of the performance of inference engines with two sub-engines. Figure 4 shows the detection results of both sub-engines along with the final pixel-level detection compared to the ground truth for a single NIR hyperspectral image with 5 × 5 mm FMs.

Overall, the transition from CPU to GPU platforms, along with the introduction of hardware acceleration and model compression, has had little effect on accuracy. Across different inference engines, the precision, recall, F1 scores, BACC, and ACC remained stable and nearly identical, with only three cases with a negligible difference of 0.1%. Notably, F1 scores consistently exceeded 84.1% for the muscle channel sub-engine and 87.2% for the non-muscle channel sub-engine across all inference engines. The precision and recall of the TensorRT-GPU-PTQ engine were reduced by 0.1% for the muscle channel sub-engine (from 88.1% to 88%) and the non-muscle channel sub-engine (from 77.6% to 77.5%), respectively, while the BACC (88.8%) and ACC (99.6%) remained unchanged. This confirms that both hardware acceleration and model compression to FP16 effectively preserve the inference engine’s accuracy without additional model training.

The consistency in accuracy across different inference engines highlights the robustness of the inference model with two GAN discriminators and validates the effectiveness of GPU acceleration and PTQ optimization strategies. These findings provide strong evidence that these strategies can be implemented without compromising model accuracy, making them ideal for high-throughput, real-time detection applications in industrial settings.

### 3.2. Static vs. Dynamic Batch Size

We compared the performance of static and dynamic batch size configurations using TensorRT-GPU engines. This analysis is essential because it helps us decide whether we can use a TensorRT inference engine with a single dynamic batch size, which offers great flexibility without building multiple inference engines. This flexibility is particularly beneficial in high-throughput imaging scenarios common in industrial settings without compromising performance. The results, summarized in Table 2, highlight the comparative metrics for both batch-size types across various batch sizes.

We observed that both static and dynamic batch-size configurations achieved comparable performance across the measured metrics, including mean inference time, standard deviation of inference time, and throughput, at each specified batch size. At the smallest batch size of 1 KB, both configurations maintained an inference time of 1.2 ms and a throughput rate of 0.81 MB/s. As the batch size increased exponentially by powers of 10, throughput (in MB/s) also increased similarly for both batch-size types, attributed to the utilization of GPU processing capabilities. Even at larger batch sizes, such as 100 MB and 1 GB, dynamic batch-size configurations maintained performance comparable to their static counterparts.

These findings suggest that while both static and dynamic batch sizes deliver similar performance in terms of latency time and throughput, dynamic batch sizing offers significant flexibility. It can accommodate a wider range of input sizes without compromising performance efficiency. This flexibility is particularly beneficial in FM detection applications, where changes in line speeds or imaging conditions necessitate an inference engine capable of handling varying input sizes. The minimal performance trade-off, combined with the added flexibility, makes the dynamic batch size configuration preferable for high-throughput, real-time detection environments. As a result, the dynamic batch size became the selected option for the subsequent efficiency analyses in Section 3.3.

### 3.3. Optimization Performance Analyses

#### 3.3.1. Latency

Latency, defined as the average inference time per batch, is a critical metric for evaluating the real-time performance of inference engines. We examined the inference latency for each sub-engine within across four different inference engines. Figure 5 illustrates latency trends as a function of batch size for both sub-engines across all four inference engines. Generally, latency increased with larger batch sizes due to increased processing requirements for more data, though the rate of increase varied among different inference engines.

Initially, the unoptimized TensorFlow-CPU and TensorFlow-GPU engines showed negligible latency increments at smaller batch sizes, with latencies ranging from 6.87 ms to 16.86 ms for both inference engines up to 100 KB. However, when the batch size was increased to the maximum batch size of 1 GB, the TensorFlow-CPU exhibited a steep increase in latency, reaching up to 6.851 s. While the TensorFlow-GPU showed improvements compared to the TensorFlow-CPU, it still showed significant latency increases, peaking at 1.391 s.

In contrast, the hardware-accelerated TensorRT-GPU showed a significant reduction in latency across all batch sizes when compared to the unoptimized inference engines, achieving up to 5 times faster performance than TensorFlow-GPU and 21 times faster than TensorFlow-CPU. At the minimal batch size of 1 KB, latency significantly decreased to 1.2 ms and remained impressively low even at the maximum batch size of 1 GB, where it recorded approximately 322 ms. The hardware acceleration and the introduction of model compression via PTQ in TensorRT-GPU-PTQ further reduced latency, especially for larger batch sizes. This optimization strategy using hardware-accelerated PTQ achieved the lowest latencies, maintaining exceptionally low latency for the largest batch size, capped at around 285 ms for both inference channels.

These findings confirm that using hardware acceleration and model compression not only preserves the quality of predictions but also enhances speed significantly, thus supporting high-efficiency operational needs in poultry processing. The results of this latency analysis suggest that the applied optimizations—particularly the use of PTQ in conjunction with GPU acceleration—provide a robust solution for maintaining throughput without sacrificing performance, marking a substantial advancement in the deployment of deep learning models in real-time detection systems.

#### 3.3.2. Throughputs

Throughput, measured in MB/s, represents the amount of data processed per second and serves as a key measure of model efficiency, particularly in high-throughput environments like poultry processing lines. We assessed the throughputs of four different inference engines for the two sub-engines utilizing muscle and non-muscle GAN discriminators at the pixel level, as shown in Figure 6.

The unoptimized TensorFlow-CPU engine showed moderate throughput at smaller batch sizes, starting at 0.13 MB/s for 1 KB. As the batch sizes increased, there was a noticeable improvement in throughput, peaking at 158.5 MB/s for a batch size of 10 MB. At the larger batch sizes of 100 MB and 1 GB, throughput decreased slightly. The unoptimized TensorFlow-GPU engine comprising two sub-engines exhibited better throughput from the beginning, showing 0.16 MB/s at a 1 KB batch and significantly rising to 703.13 MB/s at the maximum batch size of 1 GB.

In contrast, the hardware-accelerated TensorRT-GPU inference engine demonstrated a substantial enhancement in throughput across all tested batch sizes compared to the unoptimized engines. The throughput with this inference engine started with a higher throughput of 0.81 MB/s at 1 KB and achieved exceptional performance at larger batch sizes, reaching up to 2.933 GB/s at a batch size of 100 MB, and maintaining a high throughput of 3.037 GB/s at the largest batch size of 1 GB. The model compression in conjunction with GPU acceleration through TensorRT marked a significant advancement. Compared to the TensorRT-GPU, the TensorRT-GPU-PTQ maintained similar throughput rates at smaller batch sizes but exhibited superior performance at larger batch sizes. At a batch size of 100 MB, this engine achieved a throughput of 3.298 GB/s, the highest among all configurations. This peak throughput underscores the model’s ability to efficiently handle large volumes of data, which is crucial for real-time applications that require rapid processing.

Interestingly, across all tested configurations, throughput tended to converge around the 100 MB batch size (i.e., 260k pixels). This convergence suggests a level-off point where each engine reaches its processing capacity limit under the hardware constraints used in the study. The throughput results indicate that while all engines show increased efficiency with larger batch sizes, the optimized GPU-accelerated inference engines, especially those optimized with PTQ, provide significant improvements in processing speed. This demonstrates their suitability for deployment in industrial settings, where both high data volume and swift processing are crucial.

#### 3.3.3. Hypercube Rates

In this analysis, we evaluated the performance of four different inference engines based on their ability to process hyperspectral images per second, a crucial measure for real-time applications in poultry processing. This metric indicates how many hyperspectral data cubes each inference engine can process in one second, using two types of hypothetical hyperspectral cameras mentioned in Section 2.4.1. Figure 7 and Figure 8 show the processing rates for each camera across different batch sizes and inference engines with two sub-engines.

The unoptimized TensorFlow-CPU engine demonstrated limited processing capabilities, achieving only two images per second for the 256k-pixel SWIR camera and three images per second for the 108k-pixel NIR camera. The unoptimized TensorFlow-GPU engine showed improved performance, allowing the processing of seven hyperspectral images per second for the 256k-pixel SWIR camera and 16 for the 108k-pixel NIR camera at its maximum. Despite this enhancement, the performance of the 256k-pixel camera, which produces 2.4 times more data than the 108k-pixel camera, still falls short of meeting the high-speed requirements for processing approximately 5–9 hypercubes per second. This capability is essential for inspecting 4.7–8.3 halved fillets per second at 140 and 250 BPM, respectively, in poultry processing, highlighting the need for more efficient processing solutions. A significant advancement was achieved with the optimized inference engines utilizing TensorRT, which significantly increased hypercube rates when the batch size reached approximately 26k pixels (10 MB) or more. The hardware-accelerated TensorRT-GPU engine achieved processing speeds up to 31 hypercubes per second for the 256k-pixel SWIR camera and 75 hypercubes per second for the 108k-pixel NIR cameras at the largest batch size of 2.6 million pixels (1 GB). This improvement emphasizes the substantial impact of hardware-accelerated optimization on both throughput and real-time processing capabilities. Further enhancements were achieved with the integration of model compression in TensorRT-GPU-PTQ, establishing new benchmarks in processing efficiency. The TensorRT-GPU-PTQ processed over 35 hypercubes per second for the 256k-pixel SWIR camera and 84 hypercubes per second for the 108k-pixel NIR camera at the largest batch size of 1 GB, clearly surpassing the throughput requirements for real-time processing and confirming its suitability for industrial deployment. Models, especially those optimized with PTQ, provide significant improvements in processing speed. This demonstrates their suitability for deployment in industrial settings.

### 3.4. Discussion

The findings of this study suggest the significant advantages of advanced optimization techniques, specifically the combination of model compression and GPU hardware acceleration, for the FM detection application utilizing a deep learning model in poultry. Although the simulation conducted in this study remains partially theoretical, the validated computational workflows and optimizations provide a conceptual framework for potential implementation in live operational settings. The integration of these optimization technologies shows promise for substantial improvements in poultry meat safety and processing efficiency, aligning with the industry’s shift toward more automated and precise contamination detection systems. The optimization strategy with GPU-based acceleration and model compression significantly enhanced real-time hyperspectral imaging capabilities under high-speed imaging conditions. The optimized inference engine, equipped with model compression and hardware acceleration, demonstrated the ability to process up to 35 and 82 hyperspectral images per second for two hypothetical 256k-pixel SWIR and 108k-pixel NIR hyperspectral cameras, respectively. When each processed hyperspectral image covers the entire surface of a halved chicken breast fillet, these results suggest that the proposed optimized methods could monitor approximately 2000 to 5000 BPM, far exceeding the typical poultry processing line speed of around 140, peaking at 250 per minute (i.e., 280–500 halved fillets per minute). Such high performance will allow for the immediate detection and removal of FMs using a sorting machine, such as a mechanical diverter, which is critical for maintaining quality and safety standards. The experiment with dynamic batch sizes demonstrated that this approach increased the model’s adaptability and ensured flexibility, enabling it to effectively respond to varying input sizes while maintaining consistent output under diverse imaging conditions. These optimizations enhance processing speeds and can be applied to other rapid processing applications. Overall, the results indicate that the hardware-accelerated model compression can significantly advance industrial hyperspectral imaging applications in other food processing contexts that require both rapid and reliable real-time detection. However, this study has several limitations. One primary limitation is its current hardware dependence. The optimizations were heavily reliant on NVIDIA GPUs, which may limit the applicability of the performance enhancements to other hardware platforms, such as FPGAs, TPUs, or custom AI accelerators, where different frameworks or precision formats may be required. Although NVIDIA GPUs are considered an industry standard in GPU computing, we recognize the growing diversity of hardware solutions available. In principle, the same core optimizations—post-training quantization and hardware-aware model acceleration—can be adapted to alternative architectures by using equivalent vendor toolchains (e.g., Vitis AI for FPGAs or TensorFlow Lite for edge TPUs). Future research will investigate the feasibility and performance trade-offs of such implementations to broaden the accessibility and cost-effectiveness of these techniques.

Moreover, our work focused exclusively on FP16 post-training quantization (PTQ). While FP16 effectively balances memory efficiency and model accuracy, more aggressive approaches like INT8 quantization could further reduce computational overhead and storage requirements. However, this often requires careful calibration using a representative calibration dataset and, in some cases, fine-tuning to maintain the high sensitivity required by hyperspectral data, which goes beyond our study scope of optimizing the model without any re-training. In future work, we plan to investigate INT8 PTQ to evaluate its potential benefits in terms of both model size and real-time inference speed, while also exploring strategies to minimize any performance degradation due to lower precision.

It is important to note that while this study focused specifically on detecting FMs on boneless, skinless chicken breast fillets, the compression and acceleration techniques proposed in this study are largely independent of the detection model. Adapting our pipeline for other types of meat, such as pork or beef, or even plant-based products, would just need to retrain or fine-tune the underlying deep learning model using the relevant hyperspectral data. However, the proposed optimization methods, such as quantization and GPU acceleration, would still be applicable and effective.

Furthermore, while the experimental validation was robust, again, it was conducted under controlled conditions and might not fully capture the complexities or unpredictable variables present in real-world production scenarios. The current dataset was limited to the NIR spectral range of 1000–1700 nm and a restricted set of FMs for testing. This NIR wavelength range may not fully represent the real-world hyperspectral imaging conditions of chicken fillets, including potential FM contaminants and environmental factors, such as variations in lighting–material interactions, surface textures, and seasonal changes in chicken breast fillets. Future studies should include a broader variety of fillet types for training the model and include additional contaminant types for testing, ensuring greater applicability. An expanded dataset, combined with more diverse sampling conditions, could provide a better understanding of the optimized inference model’s robustness and its capacity to adapt to the challenges faced in poultry processing. Additionally, advanced data augmentation strategies that simulate variable sample surfaces and types may help bridge the gap between laboratory simulations and real-world applications.

Lastly, the inference model, which consists of two discriminators, was not executed on multiple GPUs, nor integrated as a single, fully parallelized inference engine in this study. A straightforward parallelization strategy could involve running each sub-engine on a separate GPU, which could theoretically reduce the total inference latency by half if hardware resources support concurrent execution. Alternatively, multi-stream processing or custom CUDA kernels on a single GPU might enable parallel execution of both discriminators. Nevertheless, our current serial approach still meets industrial speed requirements. Even if performance is reduced by half for a batch size of 26k pixels or more, the model exceeds typical throughput demands (as shown in Figure 7 and Figure 8). While the implementation and benchmarking of these advanced strategies are beyond the scope of this study, our latency and throughput data (Section 3.3) suggest that a parallel approach could yield additional gains in high-throughput environments. Therefore, future research may involve deploying two GPUs simultaneously or designing a single GPU-based inference engine with synchronized custom CUDA kernels for dual inference channels, thereby enhancing performance in real-world poultry processing lines.

To summarize, there are several potential future research avenues to expand upon the current findings and further advance hyperspectral imaging technology. Investigating advanced quantization techniques, particularly INT8 quantization, could lead to more compact models with reduced computational demands. However, this approach would require model tuning and calibration to maintain accuracy. Exploring computationally efficient architectures during model development could significantly reduce resource demands, making these systems more viable on constrained hardware platforms. Extending optimization efforts to include a broader range of platforms, such as FPGAs or specialized AI accelerators, could diversify the utility and application of these models across various industrial settings. Expanding the dataset to include various types of chicken fillets and FMs, as well as different imaging environments, could further validate the system’s robustness and ensure its real-world applicability. Finally, implementing these optimized models in actual processing settings and gathering feedback on their operational performance would provide invaluable insights into their practical effectiveness and highlight areas needing refinement. This real-world implementation is crucial for fine-tuning the models and ensuring they meet the stringent requirements of modern industrial applications, ultimately leading to more effective and adaptable hyperspectral imaging solutions.

## 4. Conclusions

This study has effectively demonstrated the capabilities of GPU hardware acceleration and model compression techniques in enhancing the real-time detection of FMs in poultry processing, predicting processing rates of up to 82 chicken fillets per second. Such performance well exceeds the production rate of poultry processing lines, significantly improving safety and quality control measures in the poultry industry. The integration of advanced DL technologies with hyperspectral imaging has established new benchmarks for processing speeds and accuracy, proving essential for high-throughput environments where rapid and reliable detection is paramount. The application of PTQ and GPU acceleration has successfully addressed critical challenges related to the high-dimensional data typical of hyperspectral imaging, optimizing the detection process without compromising accuracy. Furthermore, the system’s ability to handle dynamic batch sizes ensured flexibility and consistent performance across varying imaging conditions. These advancements underscore the potential of such technologies to apply to many other industrial imaging applications for monitoring safety and efficiency. Nonetheless, this work is primarily designed for the NVIDIA GPU ecosystem. Adapting the proposed compression and acceleration strategies for alternative hardware platforms, such as TPUs, FPGAs, and edge accelerators, will present a valuable opportunity for exploration. This adaptation can enhance the accessibility and cost-effectiveness of real-time hyperspectral imaging solutions. Additionally, while our experiments focused on FP16 PTQ, implementing more aggressive quantization methods such as INT8 could further improve throughputs and memory savings, provided that they are carefully calibrated. Future investigations should explore these methods while balancing trade-offs between precision, detection accuracy, and practical deployment constraints. It is also essential to conduct future studies that incorporate additional contaminants and varying imaging conditions to validate model robustness under real production scenarios. This broader validation will confirm the system’s feasibility for industrial uses and enhance its overall reliability. It is also important to note that our tests were conducted in a controlled laboratory environment. In contrast, industrial processing lines present a range of real-world variables—such as conveyor speeds of up to 250 birds per minute, mechanical vibrations, and fluctuations in temperature or humidity changes—that can affect hyperspectral imaging. Consequently, pilot-scale or commercial-level testing in the future will be crucial for assessing our system’s resilience and identifying any additional optimizations necessary to maintain high detection accuracy under operational conditions. In conclusion, this study contributes significantly to the field of real-time hyperspectral imaging in food safety, pushing the boundaries of what is possible in FM detection. It paves the way for future innovations that could further enhance the robustness and efficiency of food processing technologies, ensuring they keep pace with the demands of modern production environments and regulatory standards.

## Figures and Tables

**Figure 1 sensors-25-00970-f001:**
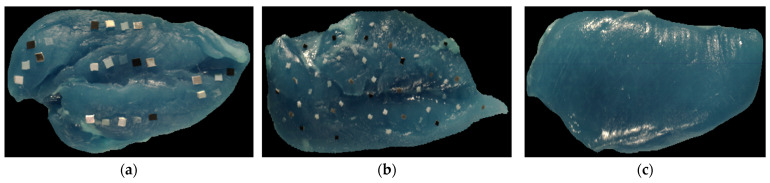
RGB pseudo-color images derived from NIR hyperspectral images with (**a**) 5 × 5 mm FMs; (**b**) 2 × 2 mm FMs; (**c**) no FMs.

**Figure 2 sensors-25-00970-f002:**
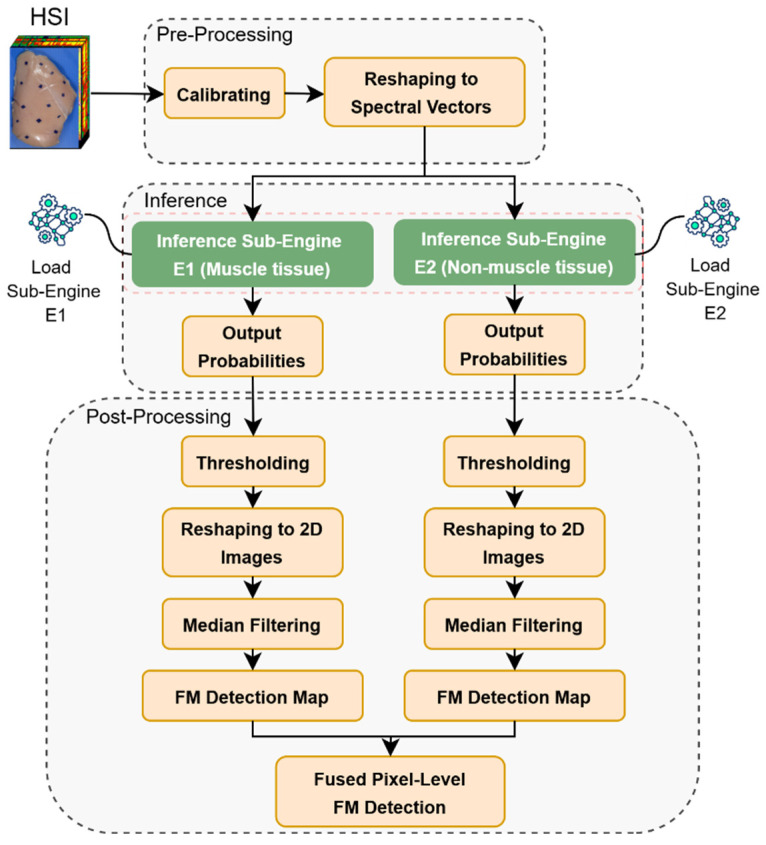
Inference pipeline for FM detection.

**Figure 3 sensors-25-00970-f003:**
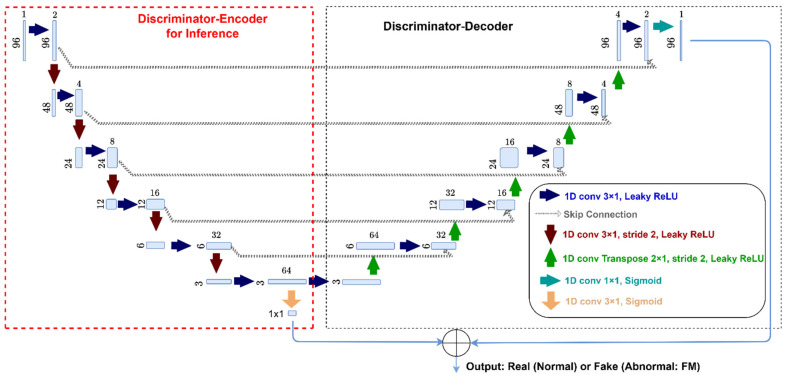
Architecture of the GAN discriminator encoder (shown in a red dotted box) for inference, as part of the overall discriminator, which includes both an encoder and a decoder used during training. Two GAN discriminators are trained to effectively differentiate between the spectral signatures of muscle and non-muscle external tissues.

**Figure 4 sensors-25-00970-f004:**
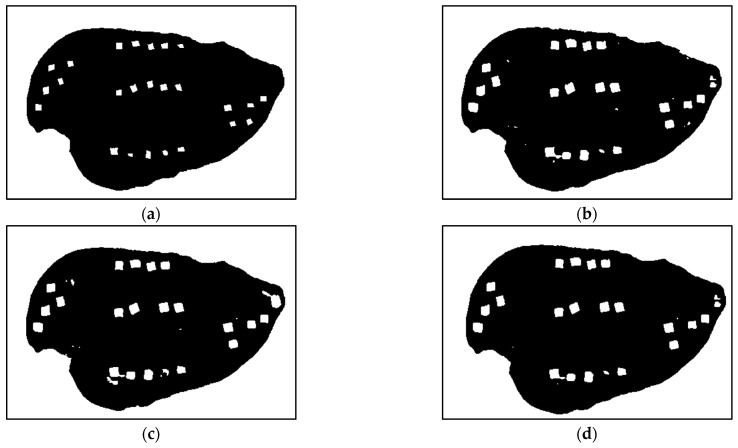
Binary detection result maps from (**a**) ground truth; (**b**) sub-engine E1 (muscle channel); (**c**) sub-engine E2 (non-muscle external tissue channel); (**d**) fused model.

**Figure 5 sensors-25-00970-f005:**
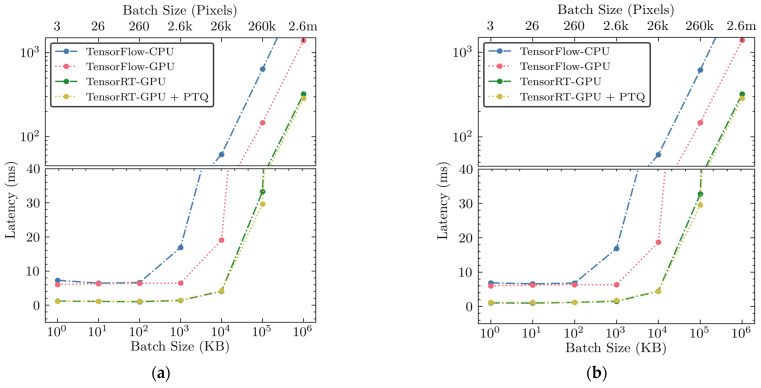
(**a**) Latency versus batch size for sub-engine E1 across all four inference engines; (**b**) latency versus batch size for sub-engine E2 across all four inference engines.

**Figure 6 sensors-25-00970-f006:**
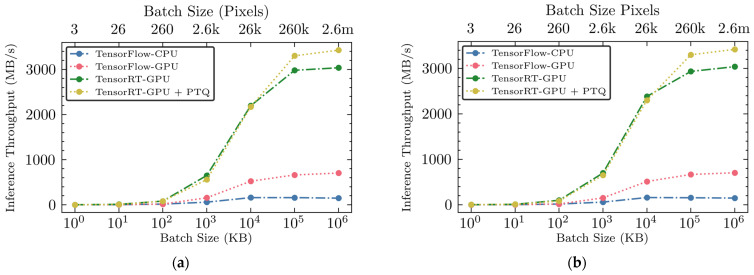
(**a**) Inference throughput versus batch size for sub-engine E1 across all four inference engines; (**b**) Iinference throughput versus batch size for sub-engine E2 across all four inference engines.

**Figure 7 sensors-25-00970-f007:**
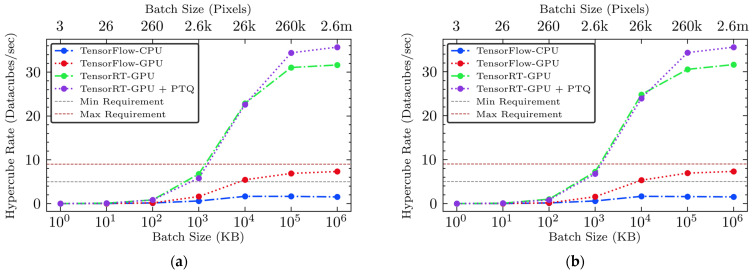
(**a**) The 256k-pixel SWIR camera hypercube rate versus batch size for sub-engine E1 (muscle discriminator) across all four inference engines; (**b**) 256k-pixel SWIR camera hypercube rate versus batch size for sub-engine E2 (non-muscle discriminator) across all four inference engines.

**Figure 8 sensors-25-00970-f008:**
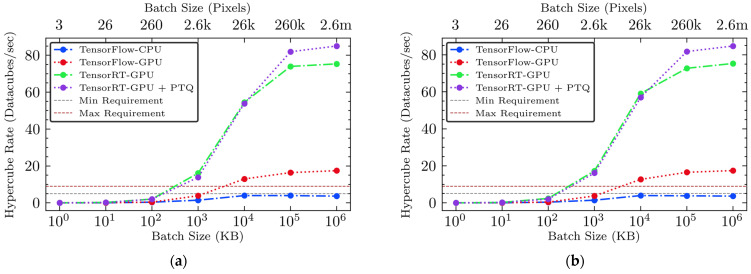
(**a**) The 108k-pixel NIR camera hypercube rate versus batch size for sub-engine E1 (utilizing muscle discriminator) across all four inference engines; (**b**) 108k-pixel NIR camera hypercube rate versus batch size for sub-engine E2 (utilizing non-muscle discriminator) across all four inference engines.

**Table 1 sensors-25-00970-t001:** FM detection accuracies of different inference engines are measured in precision, recall, F1 score, balanced accuracy (BACC), and overall accuracy (ACC).

Inference Engine	Sub-Engine	Precision	Recall	F1 Score	BACC	ACC
TensorFlow-CPU	Muscle channel only Non-muscle channel only Fused channels	0.881 0.964 0.981	0.806 0.796 0.776	0.841 0.872 0.867	0.902 0.897 0.888	0.9948 0.9959 0.9960
TensorFlow-GPU	Muscle channel only Non-muscle channel only Fused channels	0.881 0.964 0.981	0.806 0.796 0.776	0.841 0.872 0.867	0.902 0.897 0.888	0.9948 0.9959 0.9960
TensorRT-GPU	Muscle channel only Non-muscle channel only Fused channels	0.881 0.963 0.981	0.806 0.796 0.776	0.841 0.872 0.867	0.902 0.897 0.888	0.9948 0.9959 0.996
TensorRT-GPU-PTQ	Muscle channel only Non-muscle channel only Fused channels	0.880 0.964 0.981	0.806 0.795 0.776	0.841 0.872 0.867	0.902 0.897 0.888	0.9948 0.9959 0.9960

**Table 2 sensors-25-00970-t002:** Comparative performance of batch-size configurations.

Batch Size Type	Batch Size (Pixels)	Size/Batch (KB) ^1^	Mean Inference Time/Batch (ms)	SD ^2^ of Inference Time/Batch (ms)	Throughput (MB/s)
Static	3 26 260 2604 26,042 260,417 2,604,167	1 10 100 1000 10,000 100,000 1,000,000	1.2 1.09 1.1 1.4 4.84 33.17 320.63	0.42 0.32 0.31 0.51 0.83 0.84 0.93	0.81 8.88 88.62 696.51 2016.02 2944.34 3046.88
Dynamic	[1, 3, 10] ^3^ [1, 26, 50] [1, 260, 500] [1, 2600, 5000] [1, 26,042, 50,000] [1, 260,417, 500,000] [1, 2,604,167, 5,000,000]	1 10 100 1000 10,000 100,000 1,000,000	1.2 1.11 1.2 1.5 4.45 33.3 321.95	0.42 0.32 0.42 0.52 0.49 0.48 5.29	0.81 8.86 81.27 649.81 2194.73 2932.62 3037.11

^1^ The values in KB are approximated numbers. ^2^ SD: Standard deviation ^3^ [minimum, optimal, maximum] configures a range and an optimal size.

## Data Availability

The data presented in this study are available upon request from the corresponding author but are not publicly accessible without authorization from the USDA-ARS.

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
