# Peer review of "Deep Learning Model Compression and Hardware Acceleration for High-Performance Foreign Material Detection on Poultry Meat Using NIR Hyperspectral Imaging"

_sensors, 2025, doi:10.3390/s25030970_

Round 1

Reviewer 1 Report

Comments and Suggestions for Authors

This study optimizes a semi-supervised foreign materials (FM) detection deep learning (DL) model through Post-Training Quantization (PTQ) and hardware acceleration. Leveraging two pre-trained GAN discriminators and hyperspectral imaging (HIS) data, the model size is reduced by half. Simulation results indicate strong potential for real-time deployment. The paper is well-written, with clear language and credible experiments. However, it is recommended to include visualizations of the dataset and detection results for better clarity.

In summary, the reviewer suggests acceptance after minor revisions.

Reviewer 2 Report

Comments and Suggestions for Authors

The article demonstrates the application of post-training quantization (PTQ) and hardware acceleration using TensorRT to speed up inference of deep models on hyperspectral data. While these methods are well-known individually, their combined use for real-time tasks in the food industry represents a significant advancement. The article structure generally follows the format accepted by MDPI for research articles (Introduction, including a review of related work; Models and Methods; Results; Discussion; Conclusions). The English level is acceptable, making the article easy to read. The figures are of reasonable quality. The article cites 36 sources, some of which are outdated. The References section is formatted carelessly.

Comments and Recommendations on the Article:

1. The optimization methods proposed in the article are implemented exclusively on Nvidia GPUs using TensorRT, making the approach highly dependent on this hardware architecture. This limits its applicability in other environments, such as FPGAs, TPUs, or specialized processors. Considering the growing diversity of hardware solutions for deep learning, this limitation reduces the overall versatility of the research and complicates its adaptation to more cost-effective or widely available platforms.

2. The article explores only one compression strategy—post-training quantization (PTQ) to FP16. However, more aggressive approaches, such as INT8 quantization, were not investigated. While FP16 retains model accuracy, INT8 quantization could significantly reduce computational costs and memory usage but requires additional calibration and validation. The lack of experiments with INT8 leaves the question open as to how effective lighter models could be without compromising performance.

3. The authors focus primarily on empirical results and the description of optimization steps but fail to provide a detailed analytical rationale for the effectiveness of the proposed methods. For instance, the article lacks a formal analysis of the impact of quantization on computational complexity or a theoretical assessment of the acceleration achieved through hardware optimization.

4. The dataset used focuses on a limited spectral range (1000 to 1700 nm) and includes only specific types of contaminants. This approach may not account for all possible variations encountered in real-world production conditions. Factors such as lighting changes, surface texture, and other variables that could significantly affect model performance are not analyzed.

5. Experiments were conducted in a strictly controlled laboratory setting, which does not account for complexities in real production environments, such as high conveyor speeds, mechanical vibrations, or variable temperatures. This reduces the practical significance of the results.

6. The article discusses the possibility of implementing two discriminators on separate GPUs or through optimized CUDA cores, but these options were not realized. The authors note that sequential execution of sub-engines on a single GPU is a straightforward solution but do not explore how parallel implementation could enhance performance.

7. The optimization strategy targets only a single type of data (skinless chicken fillets), limiting its application to other areas, such as processing different types of meat or other food products. Each new task would require model retraining and parameter optimization.

Reviewer 3 Report

Comments and Suggestions for Authors To ensure the safety and quality of poultry products, it is necessary to effectively detect and remove foreign objects during the processing. Hyperspectral imaging (HSI) provides a non-invasive technique for distinguishing different types of (FMs) in poultry meat and fat. Due to the high dimensionality of HSI data and the computational complexity of deep learning (DL) models, analysis speed has become an obstacle to the fast-paced demands in poultry processing environments. In this study, the authors address these challenges by optimizing DL inference for HSI-based foreign material detection through a combination of PTQ and hardware acceleration techniques, and achieved good results that would contribute to improving production efficiency.   Obviously, hardware acceleration technology can definitely improve computing speed. For the PTQ technique, the author has provided relatively little introduction. Please briefly supplement the relevant principles to facilitate readers' understanding of this technique. In addition, for the fat and meat channels in the spectral imaging model, (1) does meat refer to the muscle (mainly composed of protein)? Usually, meat is composed of fat and muscle, and the author needs to accurately express it. (2) For fat, meat, and FMs, the spectral differences are significant and easy to distinguish. If the characteristic wavelengths of fat and muscle are selected and a model is established, the computational workload can be significantly reduced, which may be an efficient and feasible method.

Round 2

Reviewer 2 Report

Comments and Suggestions for Authors

I have formulated the following comments on the previous version of the article:

1. The optimization methods proposed in the article are implemented exclusively on Nvidia GPUs using TensorRT, making the approach highly dependent on this hardware architecture. This limits its applicability in other environments, such as FPGAs, TPUs, or specialized processors. Considering the growing diversity of hardware solutions for deep learning, this limitation reduces the overall versatility of the research and complicates its adaptation to more cost-effective or widely available platforms.

2. The article explores only one compression strategy—post-training quantization (PTQ) to FP16. However, more aggressive approaches, such as INT8 quantization, were not investigated. While FP16 retains model accuracy, INT8 quantization could significantly reduce computational costs and memory usage but requires additional calibration and validation. The lack of experiments with INT8 leaves the question open as to how effective lighter models could be without compromising performance.

3. The authors focus primarily on empirical results and the description of optimization steps but fail to provide a detailed analytical rationale for the effectiveness of the proposed methods. For instance, the article lacks a formal analysis of the impact of quantization on computational complexity or a theoretical assessment of the acceleration achieved through hardware optimization.

4. The dataset used focuses on a limited spectral range (1000 to 1700 nm) and includes only specific types of contaminants. This approach may not account for all possible variations encountered in real-world production conditions. Factors such as lighting changes, surface texture, and other variables that could significantly affect model performance are not analyzed.

5. Experiments were conducted in a strictly controlled laboratory setting, which does not account for complexities in real production environments, such as high conveyor speeds, mechanical vibrations, or variable temperatures. This reduces the practical significance of the results.

6. The article discusses the possibility of implementing two discriminators on separate GPUs or through optimized CUDA cores, but these options were not realized. The authors note that sequential execution of sub-engines on a single GPU is a straightforward solution but do not explore how parallel implementation could enhance performance.

7. The optimization strategy targets only a single type of data (skinless chicken fillets), limiting its application to other areas, such as processing different types of meat or other food products. Each new task would require model retraining and parameter optimization.

The authors have addressed all my comments. I found their responses quite convincing. I support the publication of the current version of the article. I wish the authors creative success.